# Experimental guidance for discovering genetic networks through hypothesis reduction on time series

**Breschine Cummins**[1]*, **Francis C. Motta**[2], **Robert C. Moseley**[3],
**Anastasia Deckard**[4], **Sophia Campione**[3], **Marcio Gameiro**[5,6], **Tomáš Gedeon**[1],
**Konstantin Mischaikow**[5], **Steven B. Haase**[3]

**1** Department of Mathematical Sciences, Montana State University, Bozeman, Montana, United States of America, **2** Department of Mathematical Sciences, Florida Atlantic University, Boca Raton, Florida, United States of America, **3** Department of Biology, Duke University, Durham, North Carolina, United States of America, **4** Geometric Data Analytics, Durham, North Carolina, United States of America, **5** Department of Mathematics, Rutgers University, New Brunswick, New Jersey, United States of America, **6** Instituto de Ciências Matemáticas e de Computação, Universidade de São Paulo, São Carlos, São Paulo, Brazil

* breschine.cummins@montana.edu

**Data Availability Statement:** The main software product introduced in the manuscript is publicly available under the MIT license in the GitLab

## Abstract

Large programs of dynamic gene expression, like cell cyles and circadian rhythms, are controlled by a relatively small "core" network of transcription factors and post-translational modifiers, working in concerted mutual regulation. Recent work suggests that system-independent, quantitative features of the dynamics of gene expression can be used to identify core regulators. We introduce an approach of iterative network hypothesis reduction from time-series data in which increasingly complex features of the dynamic expression of individual, pairs, and entire collections of genes are used to infer functional network models that can produce the observed transcriptional program. The culmination of our work is a computational pipeline, **I**terative **N**etwork **H**ypoth**e**sis **Re**ductio**n** from **T**emporal **Dynamics** (Inherent dynamics pipeline), that provides a priority listing of targets for genetic perturbation to experimentally infer network structure. We demonstrate the capability of this integrated computational pipeline on synthetic and yeast cell-cycle data.

## Author summary

In this work we discuss a method for identifying promising experimental targets for genetic network inference by leveraging different features of time series gene expression data along a chained set of previously published software tools. We aim to locate small networks that control oscillations in the genome-wide expression profile in biological functions such as the circadian rhythm and the cell cycle. We infer the most promising targets for further experimentation, emphasizing that modeling and experimentation are an essential feedback loop for confident predictions of core network structure. Our major offering is the reduction of experimental time and expense by providing targeted guidance

repository https://gitlab.com/biochron/inherent_dynamics_pipeline.git. All data and scripts used to generate the figures and results in the manuscript are publicly available under the MIT license in the GitLab repository https://gitlab.com/biochron/2022-inherent-dynamics-pipeline.git.

**Funding:** BC and TG were supported by NSF TRIPODS+X grant DMS-1839299, DARPA FA8750-17-C-0054, and NIH 5R01GM126555-01. FCM and AD were supported by DARPA FA8750-17-C-0054. SBH and RCM were supported by DARPA FA8750-17-C-0054 and NIH 5R01GM126555-01. SC was supported by NIH 5R01GM126555-01. KM and MG were partially supported by the National Science Foundation under awards DMS-1839294 and HDR TRIPODS award CCF-1934924, DARPA contract HR0011-16-2-0033, and NIH 5R01GM126555-01. KM is also supported by a grant from the Simons Foundation. MG was also partially supported by FAPESP grant 2019/06249-7 and by CNPq grant 309073/2019-7. The funders had no role in study design, data collection and analysis, decision to publish, or preparation of the manuscript.

**Competing interests:** The authors have declared that no competing interests exist.

from computational methods for the inference of oscillating core networks, particularly in novel organisms.

## 1 Introduction

Systems biologists aim to understand molecular systems comprised of gene/protein interactions. The challenge of understanding the mechanistic properties of the system stem from high-dimensional and often nonlinear interactions between genes and proteins in a network. The complexity of interactions leads to an intractably large hypothesis space that cannot be exhaustively explored by experimental approaches. Thus, there is a need for constructing computational approaches for prioritizing models that can then be interrogated by experimentalists.

Experimentally, networks have been inferred from high-throughput genomic and proteomic approaches that identify protein-protein [1–3], protein-DNA [4] or gene by gene interactions [5, 6]. Although these approaches can map interactions, they don't indicate whether the interaction is spurious or performs a specific function, and don't reveal the sign of the interaction (e.g. activation or repression). Alternatively, experimental approaches utilizing genetic manipulations such as gene knockouts or over-expression coupled with 'omics analyses of the resulting cellular responses have been used to infer functional network connections. For example, if the gene for transcription factor A is knocked out and the expression of gene B goes down, it can be inferred that A activates B [6, 7]. This inference is functional and has been used to identify clusters of co-regulated genes, but the approach lacks the capability to infer whether the regulation is direct. Although physical interaction experiments and genetic experiments have been used successfully in genetically tractable model systems with well-annotated genomes, they are expensive and time consuming. These approaches are also largely intractable for non-model systems of interest. Thus, the development of computational approaches for network inference is important.

From a computational perspective, the generic approach has been to infer local network interactions that are then assembled into the global network of interest, and then construct the models and estimate associated parameters that describe the nonlinear relationships between nodes in the networks. Ideally, these approaches utilize data that describe the entire system and is relatively easy to collect. For Gene Regulatory Networks (GRNs) that control programs of gene expression, transcriptomics measurements have been used to infer underlying network topology [8, 9]. For GRNs that control gene expression dynamics, time-series transcriptomic measurements have been used to infer both network topology and the type of interactions (activation or repression) enabling the construction of directed network graphs, e.g. [10–14]. Some methods explicitly leverage prior biological knowledge in the form of experimental evidence of interactions to improve inferences [11, 15–17]. Others refine inferences by improving coarse, global network properties of very large interaction networks such as node degrees, hierarchical structure, clustering coefficients, synchronizability, etc. The methods in [18, 19] are general and do not incorporate dynamic models of particular systems or study the ability of the proposed networks to reproduce the observed dynamics of the system they are meant to describe. Among the many GRN inference methods, very few use both prior biological knowledge and relevant dynamic models built from the topological (network) description of interactions. One such method [20] does incorporate both prior knowledge and dynamics of the inferred networks, but assumes a linear relationship between the expression of each gene and the remaining genes, requires a choice of a "known" reference network built from

experimental evidence in pathway databases, and parameter sampling from a parameter space whose dimension grows quadratically with the number of nodes in the network.

Each of these different approaches have enjoyed some limited successes, yet challenges remain. It might be expected that the different approaches could be synergistic, and compensate for the unique challenges of each method. Interestingly, it has been discovered that aggregating models leads to better predictive value than individual models alone. This concept was reported as an outcome of the DREAM challenge [21]. A similar conclusion was reached when the outputs from epidemiological models aimed at predicting the dynamics of influenza [22] and COVID-19 [23] infections were used in an ensemble forecast. The average of the outputs of multiple models has been the best predictor of true dynamics throughout the pandemic. The challenge of this approach for network inference lies in the method used to combine and weight the outputs of different methods.

Despite the capacity of current high throughput methods to produce large quantities of gene expression data, the problem of recovering an underlying GRN from experimental measurements remains under-determined, because the size of potential networks far outstrips the abundance of available data. This imbalance results in a nonidentifiability problem, in which multiple models, which may differ in both their structure and their parameterizations, can explain the observed data.

Here we describe a method for network inference that serially combines the output from computational tools into a pipeline for network inference. This pipeline is called the Inherent dynamics pipeline (**I**terative **N**etwork **H**ypothesis **Re**ductio**n** from **T**emporal **Dynamics**) [24] and is appropriate for the identification of key regulatory elements and interactions in small "core" networks that drive genome-wide oscillatory gene expression activity. As a testbed, we utilize *in silico* networks that have oscillating properties as well as experimentally verified regulatory interactions in the budding yeast cell cycle that control a large program of phase-specific gene expression as cells progress through the cell cycle. The Inherent dynamics pipeline is composed of tools that infer the set of nodes that function in the control network, the local arrangement of edges that connect the nodes, and finally the global structure and function of the network.

Rather than a mechanism for identifying the "correct" network model, we regard the Inherent dynamics pipeline as an iterative hypothesis reduction machine that transforms what would be an intractable problem, given the enormity of possible genetic controls, into a manageable set of testable hypotheses even in the absence of prior biological knowledge. Our pipeline leverages dynamic content contained in time series gene expression data in multiple ways at different stages to iteratively prune hypothesis space and ultimately produces a set of candidate networks. The prevalence of regulatory elements in the resulting set of networks provides a prioritized list of experimental interventions and the prevalence of various edges provides predictions of the impact of these interventions. We show that the Inherent dynamics pipeline is capable of providing experimental guidance for the discovery of core oscillators from gene expression time series data.

## 2 Results

### 2.1 Inherent dynamics pipeline

In principle every gene and every pairwise positive or negative regulatory interaction may be an important element in the network responsible for the oscillatory expression. This leads to an intractably large collection of hypothetical core oscillating networks. To address this challenge, our procedure is a three step framework for identifying GRNs that function as biological oscillators. An underlying assumption is that time-series gene expression data contain

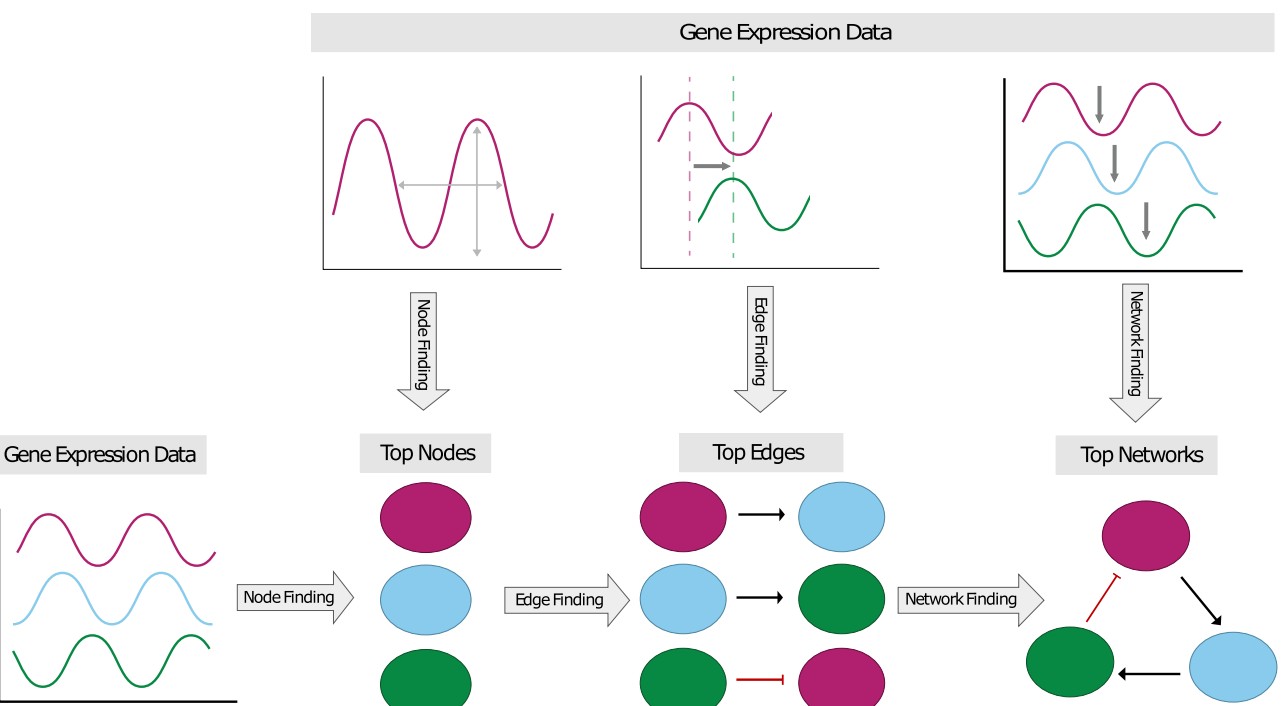

**Fig 1. Schematic of the three stages of the Inherent dynamics pipeline in which each step uses different features of the input gene expression time series data.** In the node finding step, each gene expression time trace is used independently to score the strength of periodicity. Genes with stronger periodicity and higher amplitude are hypothesized to be part of the core network and are correspondingly ranked higher. Top ranked nodes are passed into the edge finding step, where time series are used in pairs to score the likelihood of a positive or negative regulatory event in one or the other direction. A ranking is determined using high to low likelihoods and top ranked edges are passed to the network finding step, where subsets of gene expression data consisting of three or more time traces are compared to the global dynamics of network models to produce top ranking networks that are consistent with the order of peaks and troughs across the time series. Statistics of these top ranked networks are used to suggest experimental intervention at the node level.

sufficient information so that different features of the data may be used to reduce (1) the number of regulatory elements involved in producing the observed gene expression program (node finding), (2) the number of possible pairwise interactions between those regulatory elements (edge finding), and (3) the type of complex regulation occurring at each element (network finding). A schematic of the Inherent dynamics pipeline is shown in Fig 1. The focus is on identifying key regulatory components of the GRN that form a relatively small, strongly connected core network exhibiting the observed dynamics, and not on the numerous connections needed regulate all of the periodic outputs of this network.

**2.1.1 Node finding.**   Node finding is perhaps the most critical step to uncovering the gene regulatory network (GRN) that is responsible for producing the transcriptional dynamics underpinning the biological process in question, since errors made during this step will focus attention on irrelevant genes. This is a difficult task as it requires identifying the core set of genes from perhaps tens of thousands of transcribed genes. Approaches to extract a small number of core genes from tens of thousands would be experimentally time-consuming and expensive, and thus largely intractable.

The current implementation of the Inherent dynamics pipeline focuses on discovering GRNs that produce oscillatory dynamics, such as those in cell-cycle and circadian systems. The node finding step in the Inherent dynamics pipeline employs the periodicity detection algorithm DL×JTK [25]. DL×JTK uses JTK CYCLE [26] and the de Lichtenberg algorithm

**Table 1. Metrics table: Key terminology for scoring and ranking nodes, edges, and networks in the Inherent dynamics pipeline in order of computation.**

| Step | Term | Definition |
|---|---|---|
| Node finding | DL×JTK node ranking | ranking of genes according to the most periodic gene expression |
| Edge finding | local edge ranking | ranking of activating and repressing interactions according to LEM simulation |
| | top-ranked LEM edges | the top $N$ edges in the local edge ranking, a user choice |
| | local node participation score (for gene $g$) | the median rank of all edges in the top-ranked LEM edges that involve $g$ |
| | local node ranking | rank ordering of genes according to their local node participation score |
| Network finding | oscillation score (for a network) | the proportion of network behavior that permits a stable oscillation according to DSGRN |
| | pattern match score (for a network) | the proportion of stable oscillations that exhibit a DSGRN pattern match |
| | top-ranked DSGRN networks | networks with the desired oscillation and pattern match scores |
| | edge prevalence score (for edge $g \to g'$) | the proportion of top-ranked DSGRN networks that include $g \to g'$ |
| | global edge ranking | rank ordering of edges according to their edge prevalence score |
| | global node participation score (for gene $g$) | the median rank of all edges in the global edge ranking that involve $g$ |
| | global node ranking | rank ordering of genes according to their global node participation score |

[27] to quantify periodicity and amplitude as key features of gene expression profiles, see Methods Section 4.2.1. These two features have been shown to be characteristic gene expression features of core genes in GRNs that produce oscillatory dynamics [25, 28]. DL×JTK combines the quantification of periodicity and amplitude into one score, providing a ranked list of genes where the top of the list is enriched for core regulatory elements most critical to controlling oscillatory dynamics. This ranked list will be referred to as the **DL×JTK node ranking** (Table 1).

In a general framework of node finding, features besides periodicity and amplitude can be used (e.g. annotation or orthology to known nodes) to provide a ranking of the functional importance of transcribed gene products. However it is accomplished, the output of the node finding step—a small set of candidate core genes—is passed on to the edge finding step to evaluate regulatory relationships in a pairwise manner.

**2.1.2 Edge finding.** Ideally, node finding will have produced a list of candidate core genes, which are essential to produce the dynamic expression program of interest with high sensitivity and specificity. It has been shown [11], and our results confirmed that the dynamics of pairs of gene expression profiles of core regulatory elements at moderate temporal resolution contain enough information to meaningfully rank all potential interaction edges. In particular, by considering only local models of single-edge regulation of each node/target separately, it is possible with high sensitivity and specificity to identify true target/regulator pairs by ranking true edges above incorrect edges [11].

We adopt the Local Edge Machine (LEM) [11], as our method of ranking all allowable edges over a fixed node set, see Methods Section 4.2.2. In our particular case, this is the collection of nodes from the DL×JTK node ranking. LEM uses a Bayesian framework to infer a posterior probability distribution on the space of possible single-edge regulation models separately for each target node. LEM's original intent was to infer the most likely regulator and

form of regulation (activation or repression) of a given target from a list of potential regulators. However, by considering each of *N* nodes as a potential target with all *N* nodes as potential regulators with either an activating or repressing effect, LEM estimates $2N^2$ probabilities for each potential edge, and thereby provides a **local edge ranking** (Table 1). The word "local" is chosen here to refer to an inference based on data for a single target and regulator, in contrast to a "global" inference performed over all the edges of a purported GRN simultaneously. In this way, edge ranking can be used to reduce hypothesis space to the most promising region(s) of network space by effectively eliminating certain target/regulator pairs from the space of possible networks. The nodes in the top ranked LEM edges can be scored to form a rank-ordered list of regulatory elements called a **local node ranking** in which a subset of the nodes from the DL×JTK node ranking has been reordered to reflect the node participation in top-ranked edges (Table 1).

**2.1.3 Network finding.**   The network finding step accepts a ranked list of gene interactions that are ideally enriched by regulatory connections critical to the molecular process under consideration. Although DL×JTK and LEM have a strong tendency to highly rank ground truth nodes [25] and edges [11] respectively, false positives and false negatives do exist within the lists of top-ranked nodes and edges. Furthermore, even when both tools work perfectly, there is no guarantee that the top pairwise LEM interactions will produce a network of complex interactions that faithfully reproduces the observed data. The challenge of network finding is two-fold: (1) to quantify the ability of complex GRNs built from highly ranked edges to exhibit the experimental data and (2) to correct for over-ranked and under-ranked edges.

The task of network finding is complicated by the enormity of network space, which precludes exhaustive evaluation of all network models (see Methods Section 4.2.3). The top ranked gene interactions from LEM provide both a small initial network, or "seed network" (Fig 2), and a list of potential network interactions that localize the network search. Candidate networks within the allowable region are then provided numerical scores using a network finding tool set based on the software DSGRN [29, 30]. Given a network we use DSGRN to provide two numerical scores. The **oscillation score** (Table 1) indicates the proportion (with respect to parameters) of network model behavior that exhibits stable oscillations. The **pattern match score** (Table 1) indicates the proportion of the stable oscillations identified from the network model that exhibit a pattern match, i.e., the stable oscillation reproduces the periodic order of the maxima and minima seen in the gene expression time series data (discussed in Methods Section 4.2.3). The collection of top-ranked DSGRN networks according to these scores provides experimental guidance for the most promising intervention targets in the form of revised local node and edge rankings called **global node and edge rankings** determined by a global node participation score and an edge prevalence score (see Table 1 and Methods Section 4.3). The term "global" here refers to the ability of a GRN model to holistically reproduce the proper dynamics of a collection of time series. The global node and edge rankings are subsets of the local node and edge rankings that have been reordered to reflect their participation in networks that exhibit the desired dynamics (Fig 2).

## 2.2 Applications

Because it is hard to establish ground truth in biological systems, we first examined synthetic data in which the regulatory interactions of a core oscillator are known. We show that the Inherent dynamics pipeline can prioritize edges as targets for further investigation under conditions that mimic distinct experimental regimes, as well as identifying nodes that are not part of the core oscillator. The synthetic data do not include any added noise; we remark that the

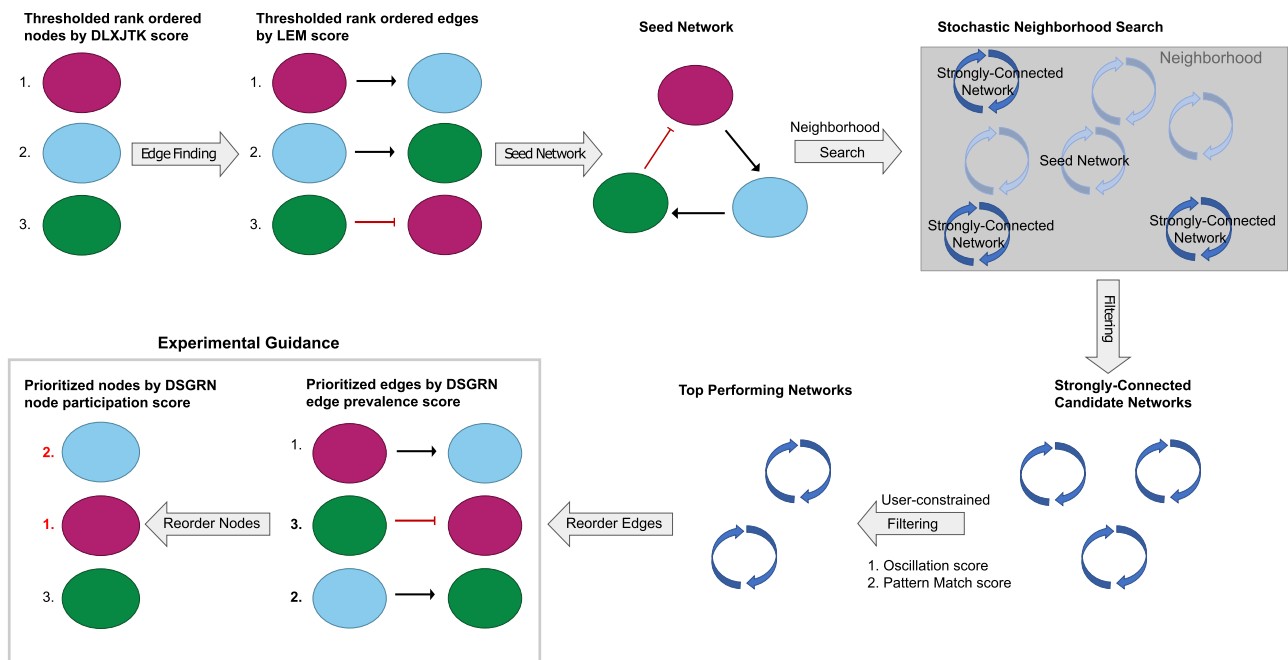

**Fig 2. Schematic of the network finding step.** From upper left in a horseshoe to lower left: The node finding step produces a thresholded list of nodes that are passed into edge finding. Edge finding ranks all possible edges between these top nodes and the very top ranked of these are used to create a seed network. The seed network is the initial condition for a neighborhood search in network space. In this neighborhood, a collection of strongly connected networks are sampled and scored according to user-specified choices of the oscillation and pattern match scores in Table 1. The participation and prevalence of nodes and edges in the top ranked networks globally matching the dynamics in the experimental data permit a reordering of nodes and edges that provides hypotheses for experimental guidance.

effects of data quality and noise have been explored previously for each module in the Inherent dynamics pipeline [11, 25, 31, 32].

We then examined the well-studied cell cycle of the budding yeast *Saccharomyces cerevisiae*. Using YEASTRACT [7], we leveraged the years of compiled experimental evidence to identify well-substantiated regulatory relationships between yeast cell-cycle genes. We demonstrated the performance of the Inherent dynamics pipeline under ideal conditions and then under conditions with decreased information availability.

In Table 1, we list important terminology for evaluating the output of the Inherent dynamics pipeline, see Methods Section 4.3 for details. Importantly, every term that is listed as a rank or median rank means that **lower** numerical scores indicate better performance. Those that are listed as proportions indicate that a **higher** numerical score is associated to better performance. The local and global edge and node rankings together are the primary metrics prioritizing experiments.

**2.2.1 Synthetic ground truth network.** We studied the performance of the Inherent dynamics pipeline on a synthetic, strongly connected regulatory network with nodes A, B, C, D, E, and F called the **ground truth network** shown in Fig 3A. Strongly connected networks are those in which there exists a path connecting each node to every other node, and thus there is at least one feedback loop between each pair of nodes in a network. This ground truth network was designed to achieve high oscillation and pattern match scores (Table 1) to mimic robust clock-like behavior. Three synthetically-generated time series, shown in Fig 3B–3D, were simulated with Hill models under widely separated parameterizations in order to produce disparate dynamical behavior, see Methods Section 4.4.1 for details. We added an

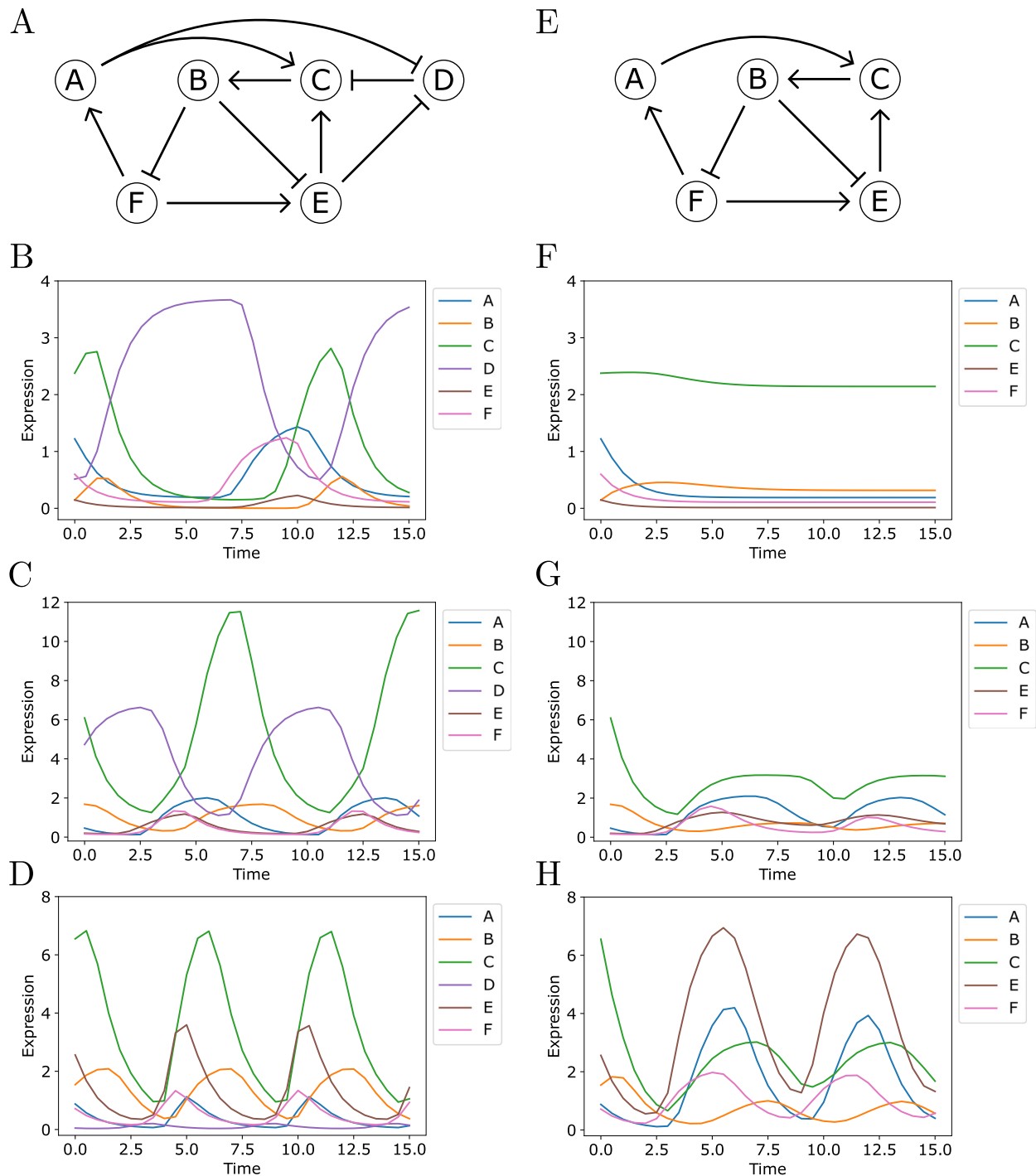

**Fig 3.** Panel A. Ground truth regulatory network, where sharp arrows indicate activation (or positive regulation) and blunt arrows indicate repression (or negative regulation). Panels B-D. Synthetic time series from 3 different parameterizations of a single Hill model (see Methods Section 4.4.1). Panel E. The subnetwork formed from the network in panel A of this figure by removing the node D. Panels F-H. Synthetic time series from the same parameters as in panels B-D of this figure, but excluding node D.

additional spurious time series, a shifted and stretched sine wave denoted G, also see Methods Section 4.4.1, which represents a node that does not participate in the simulated network. This false node G provides a negative control of the algorithm in the sense of being a "true nega-tive." Because all of the nodes in the synthetic networks are strongly connected, the node find-ing step was not needed for the synthetic data and the Inherent dynamics pipeline was run on the edge and network finding steps only.

We ran the Inherent dynamics pipeline beginning with the edge finding step for each of the three synthetically-generated time-series datasets under the hyperparameters given in Methods Section 4.4.2. Since the Inherent dynamics pipeline is stochastic, we ran five independent com-putations for every condition and report mean outcomes plus/minus one standard deviation in S1 and S2 Tables. For each dataset and each run of the Inherent dynamics pipeline, the edge finding step ranks 98 edges, which are the positive and negative edges for each pair of target/ source nodes taken from A-G. As seen previously, the top of the local edge ranking is enriched with true positives (S3, S4 and S5 Tables), consistent with the high accuracy of LEM reported in [11].

**Local inference informs building functional global network models**. Due to the high AUC scores of LEM's ranking of edges [11], we hypothesized that sampling networks in the neighborhood defined by top LEM edges put us in a region of network space that had high oscillation and pattern match scores. This claim is empirically backed by Fig 4 in which sam-pling networks at top-ranked LEM edges shows higher oscillation and pattern match scores (Fig 4A) as opposed to sampling networks from bottom-ranked LEM edges (Fig 4B). In fact, approximately half of the sampled networks, of which there are 2000 in total, exhibited a DSGRN pattern match to the observed data (see S2 Table, column 2) and therefore could not be excluded as potentially accurate models. In other words, the ability of a network model to reproduce a particular dataset was not rare in the set of networks constructed from high rank-ing LEM edges. The large number of consistent networks is a manifestation of an identifiability problem, wherein many networks of differing topologies were capable of producing the observed transcriptional oscillations. This is due to the inherent flexibility of network structure to produce different dynamics. Since a large number of network structures could explain the observed data, we do not seek to provide a "true" network to the user. Instead, we use statistics

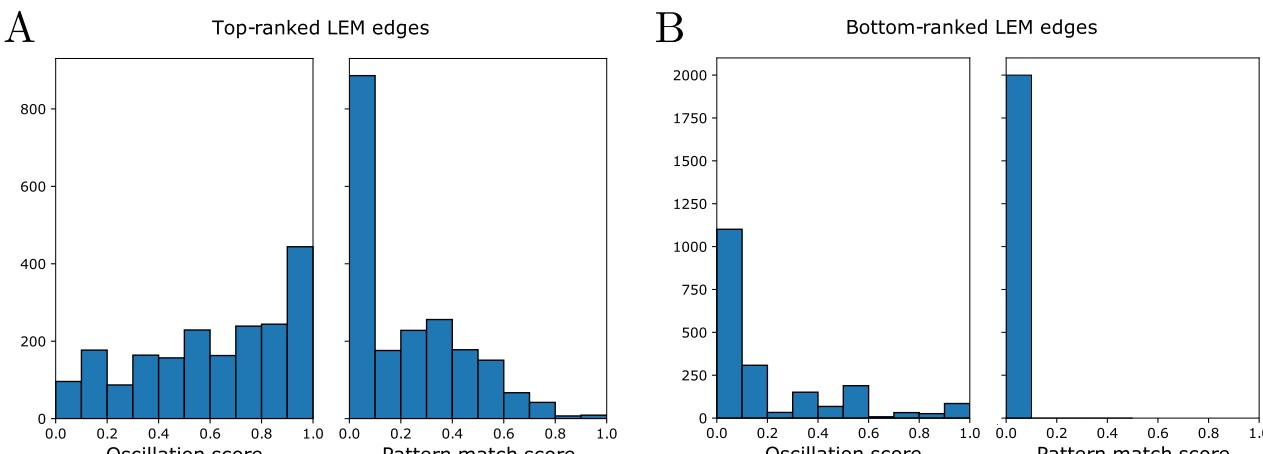

**Fig 4. Histograms of the oscillation and pattern match scores of collections of 2000 networks using in panel A, the top-ranked LEM edges of a simulation for the parameterization in Fig 3B and in panel B, the bottom-ranked LEM edges for the same edge finding step.**

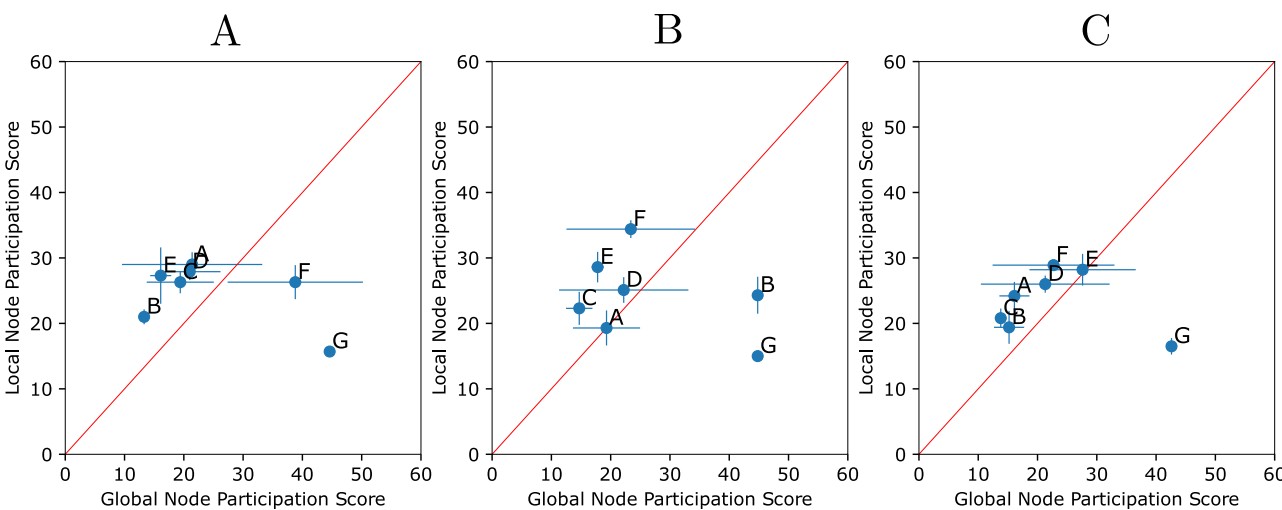

**Fig 5. Local versus global node participation scores.** Mean ± standard deviation node participation scores for the simulations shown in Fig 3B (panel A); Fig 3C (panel B); Fig 3D (panel C). Each synthetic dataset was run through the edge and network finding steps five times. The mean (blue dots) and standard deviation (blue bars) of the local node participation scores across the five runs is plotted against the mean and standard deviation of the global node participation scores. The node participation score for each simulation is computed only over the edges in the intersection of the top-ranked LEM edges of all five simulations. This excludes edges that do not have sufficiently high local edge ranks in all five simulations. Nodes located above the red diagonal line indicate an improved global node participation score versus their local node participation score. Notice that the node G is noticeably downranked in all three panels.

over the aggregate of high performing networks to pinpoint genes of interest for experimental intervention (i.e. Fig 5).

**Pattern matching can provide a large reduction in hypothesized network models.** There is uncertainty in the decay rates, binding affinities, etc. associated with a parametric network model. Therefore, networks that exhibit the desired dynamical behavior across many such parameterizations are said to exhibit the behavior more robustly. The pattern match score is a proxy for the robustness of a network model's consistency with the data by measuring the ordering of peaks and valleys in the transcriptional traces of individual genes. We reduce the stochastic sample of network hypothesis space consisting of 2000 networks when we apply oscillation and pattern match scores to assign a rank based on robustness to choose top-ranked DSGRN networks. For the synthetic data, we define top-ranked DSGRN networks as those with an oscillation score of 100% and a pattern match score of at least 50%, because the network was designed to be a robust oscillator with a fair amount of pattern matching. For an explanation of the meaning of a 100% oscillation score, see Methods Section 4.2.3. We acknowledge the correct optimization function is unknown, and this will affect our choices analyzing experimental data from the yeast cell cycle. Most of the 15 runs of the Inherent dynamics pipeline showed less than 100 networks fulfilling the criteria of a top-ranked network (see S2 Table, column 3), indicating a hypothesis reduction of an order of magnitude from the initial sample of 2000.

**Global dynamic behaviors can improve local regulatory inferences.** When removed from the context of a global network in the local inference step, a single regulatory interaction between two genes may appear highly likely due simply to spurious correlations due to limited precision in their transcriptional profiles. This improper inference is a universal problem with any inference method based on pairwise comparisons of genes. By analyzing the most likely regulatory interactions in the context of a global network model, the network finding step was able to identify false positives in the top-ranked LEM edges. The majority of the false positives

involved the true negative node G (see Methods Section 4.4.1). LEM ranked these false positives highly because all of the nodes A-F were able to effectively reproduce the sine wave G under a Hill model. On the other hand, LEM correctly identified that node G does not regulate any of nodes A-F (see S3, S4 and S5 Tables and notice that there are no edges with G as a regulator of nodes A-F with one very low-ranked exception). The network finding step then identified that node G was not able to participate in any feedback systems. This is a clear case where an analysis of the global dynamics supported by a functional network model can remove false positives that appear to be viable without the broader context of the entire network.

**The functional core oscillator may be condition-specific**. The influence a particular node has on the dynamic output of a GRN may depend on the cellular condition in which the network operates. Thus, what constitutes a core oscillator may depend on the conditions under which data were collected. We hypothesize that the different parameterizations of the ODE system that generated the data in Fig 3 are a reasonable proxy for different cellular conditions, and observe that these varying conditions can change which nodes may be justifiably called participants in a core oscillator.

A careful analysis of the results of the synthetic network shown in Fig 3D showcases how the edge rankings can identify true positive edges that do not strongly influence the global dynamics of the observed oscillations. In the analysis of the data shown in Fig 3D, the repressing edge from node D to node C never appears in the top-ranked LEM edges due to a low LEM likelihood score, unlike the results for Fig 3B and 3C. Moreover, the repressing edge from node E to node D has a zero edge prevalence score; i.e., it participates in no top-ranked DSGRN networks. This is circumstantial evidence that the node D might play a less important role in the dataset Fig 3D. We explored this phenomenon by examining the five-node subnetwork of the ground truth network shown in Fig 3E that is formed by removing the node D.

We simulated datasets at the same parameters as in Fig 3B–3D excluding node D. The results are shown in Fig 3F–3H. It is apparent that D is a critical node for oscillations in the parameter set for Fig 3B, as the removal of node D causes all oscillations to cease as shown in Fig 3F. This effect is attenuated for the parameter set for Fig 3C, which shows damped oscillations after the removal of node D. However, the removal of node D from the parameter set for Fig 3D does not halt strong oscillatory behavior, seen by comparing Fig 3D and 3H, although the quantitative values of the individual nodes are different. This presents strong evidence that the five-node subnetwork in Fig 3E can operate as the true core oscillator of the ground truth network under some parameterizations. We view these different parameterizations as proxies for distinct experimental conditions, such as different growth media or temperature. Thus the Inherent dynamics pipeline can distinguish between two strongly connected networks that operate as core oscillators under different cellular conditions. In other words, the Inherent dynamics pipeline may not return the same network structures if data are collected under different experimental conditions.

Careful consideration of the formulation and parameterizations of the ODE models that produced the time traces in Fig 3 suggests why D is a core node required for sustained oscillations in Fig 3B, but serves a diminished role in the other two parameterizations. Briefly, in the parameterization that produced Fig 3B the regulation from node D to C strongly outweighs the input from nodes A and E to node C, and this distribution of relative strength of regulation does not occur to the same degree in the parameterizations for Fig 3C or 3D. See the discussion in Methods Section 4.4.1 for more detail.

**Global dynamic behaviors can improve core variable inferences**. Our primary goal is to use the Inherent dynamics pipeline in an experiment-simulation-experiment loop that ultimately guides the discovery of core oscillator genes in non-model organisms. It is easier to perturb the expression of a gene, thereby impacting all regulatory interactions associated to the

gene, than it is to disrupt a single regulatory interaction. For this reason, we prioritize GRN nodes rather than edges as experimental targets by comparing the local and global node rankings from Table 1. This comparison yields a prioritized list of potential experimental interventions based on node participation in networks that robustly support the observed dynamic behavior.

The node participation scores for the local versus the global node rankings are seen in Fig 5. Points above the diagonal indicate nodes that are upranked in the global node ranking, while those below the diagonal are downranked. Points in the lower left corner of the diagonal are highly ranked by both LEM and DSGRN, and those in the upper right are poorly ranked by both methods.

In terms of experimental prioritization, nodes at the lower left of the diagonal should be viewed as having the greatest confidence in their participation in a core oscillator, since they rank highly at both the local and the global level. Next to be prioritized are those highest above the diagonal; i.e. those that are upranked consistently in the global node ranking. We suggest that downranked nodes be disregarded. The downranked nodes may contain false negatives; however, we observe that the true negative G is correctly identified and that the area above the diagonal is enriched with true positives, making it a more promising area of investigation. We remark that node D is not downranked on average in Fig 5C despite our finding that node D is not necessary for oscillatory behavior in the parameterized Hill model in Fig 3D and 3H. However, the wide standard deviation shows that D is downranked in some of the computational trials and, in addition, node D is upranked for the simulation in Fig 3B where we know it to be very important. In Fig 5, the most highly prioritized nodes for experimental investigation are B, E for Fig 3B; C, E for Fig 3C; and C, B, A for Fig 3D.

**2.2.2 *S. cerevisiae* cell cycle.** To further validate the inference pipeline, and examine its utility in the context of real data, we applied the Inherent dynamics pipeline to transcript expression time series collected from a *S. cerevisiae* population that was synchronized in the cell cycle. Evidence suggests that the control of periodic cell cycle transcription is largely controlled by a core GRN [33–39], and although the cell cycle of the yeast *S. cerevisiae* is very well studied, the exact topology of the core transcriptional oscillator controlling the large transcriptional program during cell division is still under investigation. However, there are experimentally substantiated interactions between known cell-cycle genes. We chose nine genes that have strong experimental evidence implicating them in the yeast cell-cycle transcriptional control, along with 24 regulatory interactions gleaned from YEASTRACT, a database that compiles experimental evidence for regulatory interactions in the yeast genome [7], along with three more edges from the cell cycle network model in [34]. See Methods Section 4.4.3 for the lists of genes and interactions. We will refer to these as **substantiated** nodes and edges. All other nodes and edges will be referred to as **unsubstantiated**.

In the yeast cell cycle, and even more so for non-model organisms, the collection of core oscillator genes is uncertain, and many non-core genes exhibit oscillatory transcriptional dynamics. To assess the performance of the Inherent dynamics pipeline, we included two "true negative" or unsubstantiated gene products, RIF1 and EDS1, that are highly oscillatory according to DL×JTK [25], but do not participate in any regulatory interaction with substantiated nodes according to YEASTRACT.

Prior biological knowledge, e.g. the identity of a core regulator, or the functional activity of a regulator as only a repressor or only an activator, could be used in principle to make *a priori* hypothesis reductions. The Inherent dynamics pipeline incorporates this information using gene annotations that record whether a given gene product acts as an activator, a repressor, or only as a target. The least constraining choice is to allow a gene product to take any of these roles. If a gene is marked as not a target, then its corresponding node in a regulatory network

will have no in-edges. Likewise, if a gene product may be neither an activator nor a repressor, then it will have no out-edges. The most interesting case is when a gene is both a regulator and a target, but is allowed to be only an activator or only a repressor. This allows the gene to be evaluated as a potential member of the core oscillator, but restricts the type of interactions that LEM will model. We call such a restriction a **nontrivial annotation**; see Methods Section 4.4.3 for the nontrivial annotations of the substantiated nodes.

We explored four scenarios representing four levels of prior biological knowledge using the Inherent dynamics pipeline, $S^+A^+$, $S^+A^-$, $S^-A^+$, and $S^-A^-$, see the table in Fig 6. $S^+$ stands for perfect knowledge of the substantiated nodes, with $S^-$ indicating that the unsubstantiated nodes EDS1 and RIF1 are assessed for participation in the core network along with the substantiated nodes. Similarly, $A^+$ indicates the presence of nontrivial annotations and $A^-$ indicates their absence. Taken together, $S^+A^+$ indicates the most *a priori* knowledge and $S^-A^-$ indicates the least. We ran the Inherent dynamics pipeline five times for each scenario under the hyperparameters given in Methods Section 4.4.3, with two replicate microarray datasets of *S. cerevisiae* wild-type transcriptomics of the yeast cell cycle [38]. Mean outcomes plus/minus one standard deviation for the five runs are shown in S6 and S7 Tables. Similar to the synthetic data case study, we see that LEM exhibits enrichment of its top ranks with substantiated edges and that generally more than half of the sampled networks are consistent with the data (S7 Table).

The scoring criteria that we use to assess top network performance is different than for the synthetic network, because a biological core oscillator does not necessarily oscillate robustly across parameter space. The cell cycle exhibits controllability in that transcriptional oscillations can be shut off at various checkpoints [33]. These non-oscillatory states of the network are present for different choices of parameters, and therefore oscillations cannot occupy the entire parameter space. For this reason, we opted for an oscillation score of 10% to 40%, with the remaining percentage of dynamical behaviors ideally including stable fixed points, imitating checkpoint behavior. We also require very robust pattern matching via a pattern match score of 100% and a requirement that both replicates must exhibit pattern matches. We emphasize that this is a user-defined choice that is based on a biological phenotype.

**Global dynamic behaviors most improve local inference when the least prior information is available**. Nontrivial annotations and high confidence core oscillator node identification are possible with model organisms, but are limited or absent for non-model organisms. We examined the performance of the Inherent dynamics pipeline with and without these two pieces of information to model several levels of prior knowledge about a core oscillator. The goal was to examine how global dynamic information affected the ranking of edges from the local inference.

We compared the performance of local and global edge rankings in Fig 6 on the subset of substantiated edges with a nonzero prevalence score for each scenario. For each simulation, the median rank of all substantiated edges that participated in at least one top-ranked DSGRN network is computed for both the local and global edge rankings. This is a measure of rank change provided that the substantiated edge was deemed important according to global dynamical behavior. Lower medians indicate upranking, or a better result. The global edge ranking upranks substantiated edges in the scenarios that include unsubstantiated nodes, enriching the top of the global edge ranking with substantiated edges. This is particularly true when we have the least information (non-trivial annotations or high confidence nodes), the $S^-A^-$ scenario, making this technique especially applicable to novel organisms or novel core oscillators. Moreover, we show that a top network in the $S^-A^-$ scenario found by Inherent dynamics pipeline is capable of reproducing the data using a standard Hill model of gene interactions (S1 File), indicating that the dynamically coarse approach taken by DSGRN informs

| Scenario | Only substantiated nodes | Nontrivial annotations |
|----------|--------------------------|------------------------|
| $S^+A^+$ | Y | Y |
| $S^+A^-$ | Y | N |
| $S^-A^+$ | N | Y |
| $S^-A^-$ | N | N |

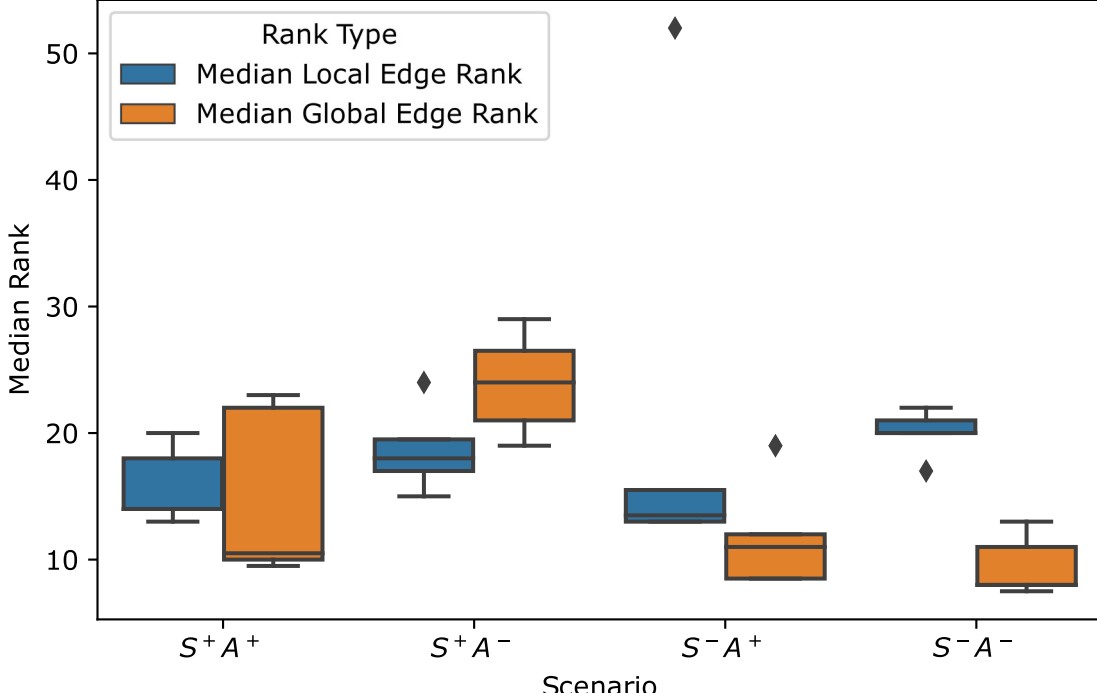

**Fig 6. Summary of yeast cell cycle rank changes for substantiated edges in global edge ranking.** The table shows the amount of information provided in each scenario. The box plots show the distributions of the median local (blue) and global (orange) edge rankings for the subset of substantiated edges with nonzero edge prevalence scores. A lower median indicates a better ranking and therefore a better result. In both $S^-$ scenarios, the global edge ranking outperforms the local edge ranking.

the parameterization of more traditional models that might provide insight into unknown cellular processes.

**Global dynamics can identify unsubstantiated nodes**. Ideally, nodes identified as promising experimental targets do not include false positives, as experimental interrogation of false positives is time-consuming and costly. We show in Fig 7 that unsubstantiated or "true negative" nodes are downranked, indicating that the Inherent dynamics pipeline does not identify them as potential core oscillator nodes. In scenarios $S^-A^+$ and $S^-A^-$, the unsubstantiated nodes EDS1 and RIF1 are present. They are downranked in the global node ranking except for RIF1 in $S^-A^+$, which is poorly ranked by both LEM and DSGRN. This finding is an indication that the network finding step improved upon the edge finding step by depressing the ranks of nodes that are unimportant to the core oscillator. Note the impact that losing nontrivial annotations can have in producing false negatives. In particular, YOX1 is downranked substantially by the global node participation score in the $S^-A^-$ scenario, but upranked in $S^-A^+$.

**Global dynamics can provide hypotheses for the functional roles of substantiated nodes**. An interesting observation from Fig 7 is that the node CLN3 is downranked in the

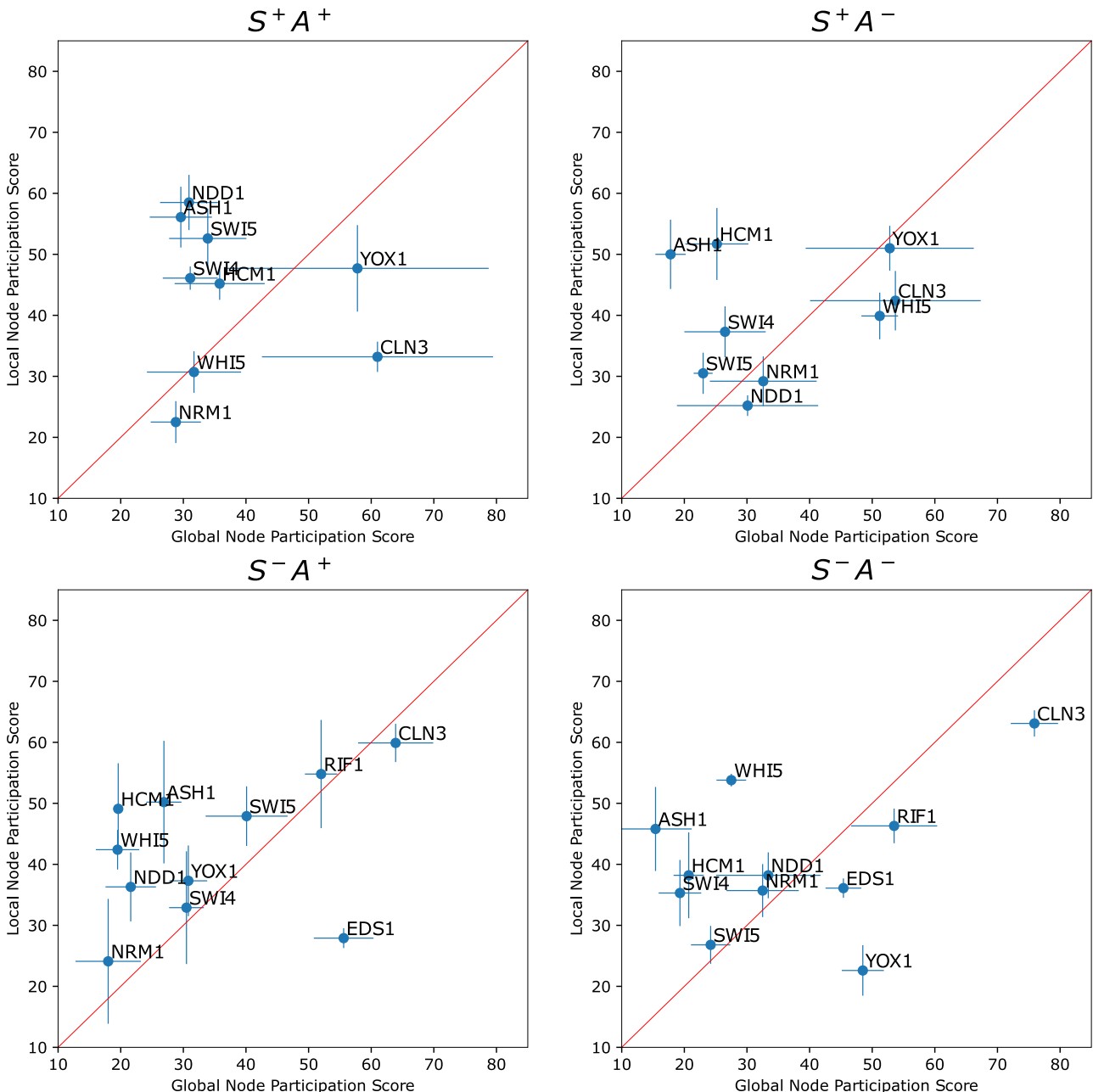

**Fig 7. Local versus global node participation scores for the yeast cell cycle network.** Mean ± standard deviation node participation scores for the four scenarios $S^+A^+$, $S^+A^-$, $S^-A^+$, and $S^-A^-$. Each scenario was run through the edge and network finding steps five times. The mean (blue dots) and standard deviation (blue bars) of the local node participation scores across the five runs is plotted against the mean and standard deviation of the global node participation scores. The node participation score for each simulation is computed only over the edges in the intersection of the top-ranked LEM edges of all five simulations. This excludes edges that do not have sufficiently high local edge ranks in all five simulations. Nodes located above the red diagonal line indicate an improved global node participation score versus their local node participation score.

global node ranking in all four scenarios, usually strongly, and sometimes is also poorly ranked in the local node ranking. We remark that CLN3 is the only substantiated node that is not a transcription factor. CLN3 is a cyclin that activates cyclin-dependent kinase (CDK), which in turn regulates the activity or stability of other proteins via phosphorylation.

There are two potential explanations for the strong down ranking of CLN3. One is that CLN3 does not play a very important role in the core oscillator. Consistent with this possibility, CLN3 is dispensable for cell-cycle progression and transcriptional oscillations [40, 41] as dilution of Whi5 by cell growth is sufficient for activating the transcriptional wave at START [42]. The other explanation is that DSGRN is most effective for transcriptional regulation rather than protein activity regulation. The work to extend DSGRN to model various types of post-transcriptional regulation such as phosphorylation is underway [43].

## 3 Discussion

When inferring GRNs from data, the space of potential core nodes, core interactions, and core networks is too large to exhaustively explore, even computationally, much less through experimentation. We demonstrate that high-throughput experimental data can be leveraged by using the software tool Inherent dynamics pipeline [24] to iteratively reduce these spaces and provide experimental guidance. We show the efficacy of this method on a synthetic network designed to exhibit robust oscillations and on yeast cell cycle data that displays controllable oscillations and a large body of experimental evidence for a particular network topology.

The Inherent dynamics pipeline network discovery tool consists of a node finding step implemented with DL×JTK [25], an edge finding step implemented with the Local Edge Machine (LEM) [11], and a network finding step dependent on the Python package Dynamic Signatures Generated by Regulatory Networks (DSGRN) [44]. The software is an iterative hypothesis reduction machine to identify core oscillators driving large scale oscillations in gene expression. The performance of the node finding step is well-documented in [25]; in this manuscript, we *a priori* identify groups of high confidence vs low confidence nodes in order to examine the performance of the combination of edge and network finding steps of the Inherent dynamics pipeline.

A notable feature of the Inherent dynamics pipeline is the synergism between the local edge finding and global network finding steps. Inference based on pairwise interactions provides an essential step to begin the search of networks by positioning the network sampler in a region of network space where networks tend to robustly reproduce the observed data. However pairwise interactions alone are insufficient to identify the dynamic function of the core regulatory network. The network finding step applies a corrective factor to the output of the edge finding step by successfully identifying false positive nodes and edges that do not participate in the core oscillator (Figs 5 and 7).

For the synthetic data, top-ranked networks were required to have an oscillation score of 100% to represent robustness, while for the yeast data, the oscillation score was limited to the range 10–40% to account for phenotypic plasticity of the cell cycle network. The fact that in addition to oscillatory behavior the network exhibits steady state behavior in the conditions that trigger one of its checkpoints highlights a difficulty in optimization of any kind for discovering networks. The mixture of cell plasticity in phenotype versus robust expression of a behavior is unknown, and therefore the classification of networks as "top performers" is uncertain. Moreover, evolution dictates that a cell only requires a sufficiently good solution, not the best solution, that is achievable under unknown developmental constraints. These constraints impact the collection of networks over which evolutionary optimization can occur, which will be highly limited with respect to all of network space. These uncertainties speak to the necessity of incorporating as much biological knowledge as possible in addition to time series data in order to increase the chances of discovering the true molecular interaction network.

The fact that the greatest gains in edge ranking by the Inherent dynamics pipeline come in situations where annotation information is the most sparse (Fig 6) suggest that the Inherent

dynamics pipeline is especially applicable to non-model organisms. However, without annotations errors may be introduced in the node finding step that will be propagated through the rest of the pipeline. Thus, improving the accuracy of the node-finding step will be a focus of future research. It has been shown that identifying core regulators can be greatly improved if genes are accurately identified as transcription factors or not [25], so improvements to computational methods, including machine learning models, for inferring gene function from readily available data (e.g., protein sequence) are desirable.

The Inherent dynamics pipeline is not proposed as a method to correctly infer as many known regulatory relationships as possible, which is the goal of many DREAM challenges [17], and the goal of many inference methods [14–16]. Rather, the approach presented here aims to identify the ostensibly small collection of core regulatory elements driving the dynamics of the much larger program. Moreover, we do not view the top ranked network or networks as the ultimate outcome of our software because of the identifiability problem wherein many models are capable of producing the same results under different parameterizations; experimental evidence is required to distinguish between the possibilities. We suggest that statistics of the top ranked networks be used to provide a prioritization of experimental interventions at the node level by re-ranking nodes according to their prevalence in dynamical network models consistent with experimental data. We demonstrate that the Inherent dynamics pipeline downranks true negative and unsubstantiated nodes in synthetic data and yeast cell-cycle data, respectively. Thus the Inherent dynamics pipeline is an appropriate tool for identifying promising experimental targets for elucidating the gene regulatory networks behind clock-like cellular phenotypes.

Inferring causation requires perturbation experiments and thus the Inherent dynamics pipeline can be utilized iteratively with experimentation. The identifiability problem means that it is hard to predict the outcome of a perturbation experiment, given that there are many network/ parameter combinations that would reproduce the data. Importantly, the Inherent dynamics pipeline can be iteratively deployed after the next round of experiments; i.e. edges that are known to exist can be enforced, new annotations can be added, and different behaviors under distinct experimental conditions can be used to further constrain the dynamic phenotype.

## 4 Methods

### 4.1 Parameters

There are several levels of parameterizations that occur in the Inherent dynamics pipeline. At one level, there is the traditional parameterization of ordinary differential equation (ODE) models with real values. The parameters for these ODE models will simply be referred to as "parameters." The parameter space for a switching system ODE is decomposed by DSGRN into a finite number of regions [44] and DSGRN computations are performed over these regions rather than over individual real values. Each such region is called a DSGRN parameter in previous publications, but we will use the term "DSGRN parameter region" in this work for clarity. Lastly, there are user choices for controlling the behavior of the numerical methods DL×JTK, LEM, and DSGRN pattern matching in the Inherent dynamics pipeline. These will be referred to as "hyperparameters."

### 4.2 Pipeline components

The Inherent dynamics pipeline [24] is a unified collection of time-series analysis algorithms tied together by data processing routines. The input to the Inherent dynamics pipeline is one or more replicate time series datasets along with a hyperparameter specification file documented in the Inherent dynamics pipeline README. To maximize platform compatibility and to ensure broad usability of the individual pieces as well as the Inherent dynamics pipeline

as a whole, we have modified or entirely rewritten each component algorithm in the Python programming language [45] and created a single Python module for installing and running the pipeline components. In addition, there is an Inherent Dynamics Visualizer (IDV) [46] that uses web-based technologies for easier interaction with Inherent dynamics pipeline output. The IDV allows the user to visualize and explore the intermediate output of each of the node, edge, and network finding steps to infer the impact of various hyperparameter choices. This facilities the incorporation of domain-specific knowledge and permits intuitive decision-making based on visual information.

**4.2.1 DL×JTK.**   The DL×JTK algorithm [25] adopts the same formulation for scoring genes as was originally defined in [27] but by combining the periodicity measure of the JTK CYCLE algorithm [26] with the regulator measure of the de Lichtenburg algorithm defined in [27]. In particular, for each gene expression profile, an empirical and an analytical $p$-value, which respectively estimate probabilities that the observed amplitude variability and the observed periodicity of the expression profile occurred at random, are first computed and then combined in a manner which accentuates expression profiles that are simultaneously highly periodic and highly variable in amplitude. Explicitly, let $G \in \mathcal{G}$ be the gene expression profile corresponding to gene $G$ in the set of all measured gene expression profiles, $\mathcal{G}$ and let $n_r$ be a positive integer. Then

$$\mathrm{DL} \times \mathrm{JTK}(G, n_r) := p_{\mathrm{reg}}(G, n_r) p_{\mathrm{per}}(G) \left[ 1 + \left( \frac{p_{\mathrm{reg}}(G, n_r)}{0.001} \right)^2 \right] \left[ 1 + \left( \frac{p_{\mathrm{per}}(G)}{0.001} \right)^2 \right]. \tag{1}$$

First to each gene is associated its so-called "regulator score", which is taken to be the standard deviation of the base 10 logarithm of the mean-normalized expression profile. In this way, the regulator score of a gene captures the deviation of the time series about its mean with a small value indicating little variation in expression from the mean expression over time. The empirical $p$-value $p_{\mathrm{reg}}(G, n_r)$ is then defined to be the fraction of $n_r$ random curves whose regulator score exceeds the regulator score of $G$ where random curves are generated by selecting at each time point the expression at that time of a curve selected uniformly from $\mathcal{G}$.

The analytic $p$-value $p_{\mathrm{per}}(G)$ is taken to be the $p$-value determined by the JTK-CYCLE periodicity scoring algorithm [26]. First, sinusoidal template curves are generated with user-specified periods and at various phase shifts determined by the sampling times of the expression profiles. A pattern of "ups" and "downs" is computed by comparing the expression level at each time with all subsequent times for both the expression curve $G$ and the equivalently sampled sinusoidal template curves. Then the total number of agreements (concordancies) and disagreements (discordancies) in the up-down pattern of $G$ and the that of the known periodic curves are computed, giving the Kendall rank correlation coefficient between the curves. By precomputing the exact null distribution of Kendalls tau correlation [47] using the Harding algorithm [48], an exact Bonferroni-adjusted $p$-value is rapidly computed for each gene. For this work, an implementation of JTK-CYCLE in Python by Alan Hutchinson [49] was modified.

**4.2.2 LEM.**   The Local Edge Machine (LEM) algorithm [11] adopts a Bayesian framework to perform inference of functional gene regulation. Namely, a prior distribution on the space of single-edge regulatory models of a given gene $G$ is updated by the conditional likelihoods that the observed expression data of gene $G$ was produced by functional regulation by gene $H$. Thus a prior distribution is first placed on a predefined set of potential regulatory models, where each model is of the standard form of a Hill function [50]:

$$\frac{dG}{dt} := \gamma - \beta G + \mathcal{F}(H), \tag{2}$$

with either a model of repression of $G$ by $H$,

$$\mathcal{F}(H) \coloneqq \mathrm{rep}(H) \coloneqq \alpha \frac{k^n}{k^n + H^n},$$

or a model of activation of $G$ by $H$

$$\mathcal{F}(H) \coloneqq \mathrm{act}(H) \coloneqq \alpha \frac{H^n}{k^n + H^n}.$$

The likelihood of the observed data given a model of regulation is estimated using the Laplace approximation formula [51] to integrate a measure of model goodness-of-fit over the five-dimensional parameter space, $(\alpha, \beta, \gamma, k, n)$. The resulting formulation explicitly balances the model error at an optimal choice of model parameters, found by a parameter optimization procedure, against the robustness of this error to small perturbations of the model parameters. Using Bayes formula, the likelihood of each regulatory model is used to update the prior distribution and produce a posterior distribution on the space of single-edge regulatory models for gene $G$. We refer to the posterior probabilities on each local model of regulation for a fixed target as the model's pld score.

In principle, the allowable model space may be expanded to include complex regulation of $G$ or models with other functional forms, but the current implementation is restricted to single-edge regulation. In the absence of any prior knowledge about gene function, the uniform distribution should be adopted as the prior distribution. On the other hand, the prior distribution on the space of allowable regulatory models may be informed by existing evidence of regulatory interactions between gene products or by known function, e.g. if there is evidence that a gene acts only as a repressor. Moreover, data from replicate experiments may be utilized by iteratively updating an initial prior. In particular, data from replicate 1 of an experiment can be used to produce a posterior distribution on model space, which is then taken to be the prior distribution on model space for data from replicate 2. This iterative posterior calculation has been included in a new implementation of the LEM algorithm that was written in Python to further improve platform compatibility, algorithm extensibility and efficiency.

**4.2.3 DSGRN pattern matching.**   DSGRN (Dynamic Signatures Generated by Regulatory Networks) [29, 30] is a software tool that, given a genetic regulatory network (GRN), creates a database of all possible dynamical behaviors that the GRN can exhibit. A GRN is represented by its nodes and interaction structure showing activating and repressing regulatory interactions between genes and gene products. This includes algebraic expressions for combining multiple input edges at target nodes, but it does not require explicit knowledge of real-valued parameters such as binding strength or decay rate. Imposing such a set of real values on a network can potentially induce qualitatively different types of dynamics.

The mathematical foundations for DSGRN [52–55] defines a general framework in which the characterization of long-term dynamical behaviors that a network can exhibit is finite. The DSGRN software identifies dynamics via these characterizations [44, 56]. DSGRN decomposes high-dimensional parameter space into a finite number of regions, where each DSGRN parameter region contains meaningful dynamical information that is true for all real-valued parameter sets inside that region. The dynamical behaviors are encoded as **state transition graphs (STGs)**, where the nodes of an STG are qualitative concentration levels of gene product, e.g. low, medium, high. The edges are permitted transitions between these system states. For example, if a repressor is currently at a high level, then its downstream target would not, unless also impacted by an activator, be permitted to increase. An STG tracks where each gene product is increasing or decreasing in concentration and where each gene product can achieve a (local) maximum or minimum expression level. A consequence is that the potential for oscillatory

behavior can be identified from the STG, as well as the stability of the oscillations and the order of the maxima and minima of different gene product concentrations within the oscillations.

The oscillation score introduced in Table 1 quantifies the amount of stable cycling exhibited in the STGs across all essential DSGRN parameter regions. It is possible, and indeed reasonably common, for a network to exhibit an oscillation score of 100%, which may appear to imply that all parameterizations of an ODE model of the network should stably cycle. The actual implication is more nuanced. While an oscillation score of 100% does not guarantee that a corresponding Hill model oscillates for all parameter selections, it strongly suggests that an ODE Hill function model with a sufficiently high Hill coefficient and initial condition in the corresponding domain of the phase space will either oscillate or exhibit decaying oscillations. These two outcomes may be difficult to distinguish experimentally.

We propose that ordering the extrema in a time series dataset is an appropriate description of the observed dynamical behavior, taking noise into account. This representation is coarse, but it qualitatively captures certain characteristics, such as relative frequency and phase differences. Given any collection of gene products, we can transform the associated time series dataset into a graph called the **data graph** where the nodes are the extrema of the time series and the edges represent events that have a known order in time [32]. Not every pair of nodes representing gene products will have an edge between them. It is possible, and in fact common, that two extrema from different time series occur close enough together that their timing is indistinguishable under an assumption of small noise.

The data graph encodes information about the procession of extrema in a similar way to that of the state transition graph. The process of attempting to match up the ordering of the extrema in the data graph and the extrema in the state transition graph is called **DSGRN pattern matching** [57]. If a pattern match exists, then we say the model is consistent with the data. When consistency exists, then the network model that produced the STG cannot be rejected as a hypothesis for explaining the experimental observations. The proportion of STGs for a network that exhibit a pattern match in a stable oscillation is the pattern match score discussed in the text.

There are limitations to the networks for which DSGRN computations are possible. The main challenge is that the combinatorial growth of the number of DSGRN parameter regions with the number of regulators into a single node can cause computations to become prohibitively expensive. To illustrate the impact of increasing network complexity, consider a simple loop of 4 nodes, where each node has one in-edge and one out-edge. This network has 12 DSGRN parameters. Now, to one node (say *A*), add three in-edges, i.e., make each of the remaining three nodes (including *A*) a regulator of *A*. This change results in nearly 3 million DSGRN parameters. Practically speaking, some empirical exploration is needed to gauge an acceptable level of network complexity, without exceeding about 4 or 5 regulators at a single node. In addition, the expense of pattern matching further reduces the size of computationally accessible networks. The user must gauge computational resources against network sample size and choose the maximum number of allowed DSGRN parameter regions accordingly. In addition, the DSGRN pattern matching technique is not yet available for self-repressing edges, although this is expected to become available in the near future.

One further limitation of the network finding step is the size of network space, which precludes exhaustive evaluation and requires sparse sampling. This is because the number of potential regulatory network structures becomes intractably large for combinations of about three nodes or more. For example, given 10 possible gene products, there are 120 choices of 3-node combinations, and for each combination, there are over 14,000 possible network structures with different combinations of positive and negative regulatory interactions [58]. This leads to well over a million possibilities for a 3-node core network even when the number of possible core nodes has been dramatically reduced. To appreciate the rate of growth, consider

that the analogous case for 4-node networks. There are 210 node combinations with over 42 million possible network structures for each combination, leading to billions of possibilities. For this reason, any network inference technique that evaluates network behavior as a whole, including ours, is required to sample network space, rather than using an exhaustive technique as in node finding and edge finding.

## 4.3 Component integration and evaluation

The DL×JTK node ranking prioritizes gene time series for further analysis in the Inherent dynamics pipeline. LEM ingests the top-ranked nodes and produces a ranked-ordered list of edges called the local edge ranking that is used to initiate the network finding step. The procedure for the network finding step is to pick a **seed network** composed of the top few LEM edges, and then to stochastically search in a neighborhood around the seed network for strongly connected networks that are sampled using a larger portion of the local edge ranking. We call edges permitted in the construction of networks the top-ranked LEM edges. The increased permissivity allows the possibility of including network edges that may have been downranked by LEM due to either stochastic computation or experimental noise. Self-repressing edges are currently removed from the top-ranked LEM edges for technical reasons, but this functionality is expected to be added in the near future.

Using the top-ranked LEM edges, the network finding step produces a sample of candidate networks in the neighborhood around the seed network. User-supplied scoring constraints are then employed to identify a collection of top performing networks. These scoring constraints are based on the fact that the number of DSGRN parameters is finite, which allows proportions of DSGRN parameter space with the desired dynamical behavior to be computed. In this work, there are two numerical scores that we use to choose top regulatory networks. The first is the proportion of DSGRN parameters that exhibit stable oscillations (**oscillation score**). The second is the proportion of stably oscillating DSGRN parameters that exhibit a pattern match to at least one dataset within the stable oscillation (**pattern match score**). When we have replicate experimental datasets, as we do for the *S. cerevisiae* data, we also require at least one pattern matching success for each replicate. Candidate networks that meet the chosen criteria are called the top-ranked DSGRN networks.

A rank-ordered list of edges called the global edge ranking is created by measuring the participation of each edge in the top-ranked DSGRN networks. Every edge is assigned an **edge prevalence score** that is the proportion of top-ranked DSGRN networks in which it appears, i.e., for the edge $i \to j$, the edge prevalence score $\mathcal{P}_{i \to j}$ is

$$\mathcal{P}_{i \to j} = \frac{N_{i \to j}}{T}, \tag{3}$$

where $T$ is the number of top networks and $N_{i \to j}$ is the number of top networks that have the edge $i \to j$ (or $i \dashv j$ for a repressing edge). $\mathcal{P}_{i \to j}$ is nonzero for any edge $i \to j$ that participates in at least one network that can faithfully reproduce the observed data to the requested degree of robustness and accuracy.

The edge prevalence score defines the global edge ranking, which is a re-ranking of the top-ranked LEM edges according to their ability to participate in complex networks with a desired phenotypic behavior. When ties in the edge prevalence score exist, they are broken by local edge rank. Any edge with a zero prevalence score is given the worst possible ranking: the number of top-ranked LEM edges.

In addition to ranking edges, we can also revise the DL×JTK node ranking for experimental prioritization. The method is the same for either the local edge ranking from LEM or the global

edge ranking from DSGRN. We use the (local or global) **node participation score**, which is computed for each node $g$ by collecting the ranks of all edges either coming into $g$ or emanating from $g$ and taking the median of these ranks. The global edge and node rankings together provide guidance to the experimentalist desiring to prioritize experiments.

## 4.4 Computational details

Data and scripts used to create images and generate statistics in Results Section 2 are located in [59].

**4.4.1 Synthetic network construction.** We constructed a collection of six-node network topologies that robustly exhibit oscillations across DSGRN parameter regions, and then we chose the top ranked of these with nontrivial regulation at a node (see node C in Fig 3A). We generated synthetic time series data from this network from three different DSGRN parameter regions. This was accomplished by sampling the DSGRN parameter regions for explicit real-valued parameters with which to simulate Hill function ODE models of the network. The simulations were evaluated for robust periodicity exhibiting a minimum of a four-fold change between peak and trough for all six synthetic genes. The three simulated datasets show distinct patterns of maxima and minima, i.e. distinct dynamical behaviors. This is roughly analogous to studying a regulatory network under three distinct experimental conditions, where each experimental condition is viewed as a different set of parameters imposed on the ODE system for the regulatory network in Fig 3A. The network's oscillation score is 100% and the pattern match scores for each time series dataset ranged from 45.8% to 51.2%.

We then added a spurious time series, "node" G, to the dataset to evaluate the performance of the Inherent dynamics pipeline with imperfect data. The time series for G was generated by

$$G = 2\left(\sin\left(\frac{9}{2\pi}t\right) + 1\right),$$

where $t$ is the vector of time points used to simulate the synthetic data, see Fig 8.

To generate the synthetic gene expression profiles from the network topologies given in Fig 3A and 3E, we simulated systems of ODEs with Hill function nonlinearities as specified in Eqs 4 and 5 respectively, with parameters given in Table 2.

$$\dot{A} = -A + \Gamma_{F,A} + \Omega_{F,A}\frac{F^n}{F^n + \theta_{F,A}^n}$$

$$\dot{B} = -B + \Gamma_{C,B} + \Omega_{C,B}\frac{C^n}{C^n + \theta_{C,B}^n}$$

$$\dot{C} = -C + \left(\Gamma_{A,C} + \Omega_{A,C}\frac{A^n}{A^n + \theta_{A,C}^n} + \Gamma_{E,C} + \Omega_{E,C}\frac{E^n}{E^n + \theta_{E,C}^n}\right)\left(\Gamma_{D,C} + \Omega_{D,C}\frac{\theta_{D,C}^n}{C^n + \theta_{D,C}^n}\right)$$

$$\dot{D} = -D + \left(\Gamma_{A,D} + \Omega_{A,D}\frac{\theta_{A,D}^n}{A^n + \theta_{A,D}^n}\right)\left(\Gamma_{E,D} + \Omega_{E,D}\frac{\theta_{E,D}^n}{E^n + \theta_{E,D}^n}\right)$$

$$\dot{E} = -E + \left(\Gamma_{B,E} + \Omega_{B,E}\frac{\theta_{B,E}^n}{B^n + \theta_{B,E}^n}\right)\left(\Gamma_{F,E} + \Omega_{F,E}\frac{F^n}{F^n + \theta_{F,E}^n}\right)$$

$$\dot{F} = -F + \left(\Gamma_{B,F} + \Omega_{B,F}\frac{\theta_{B,F}^n}{B^n + \theta_{B,F}^n}\right)$$

(4)

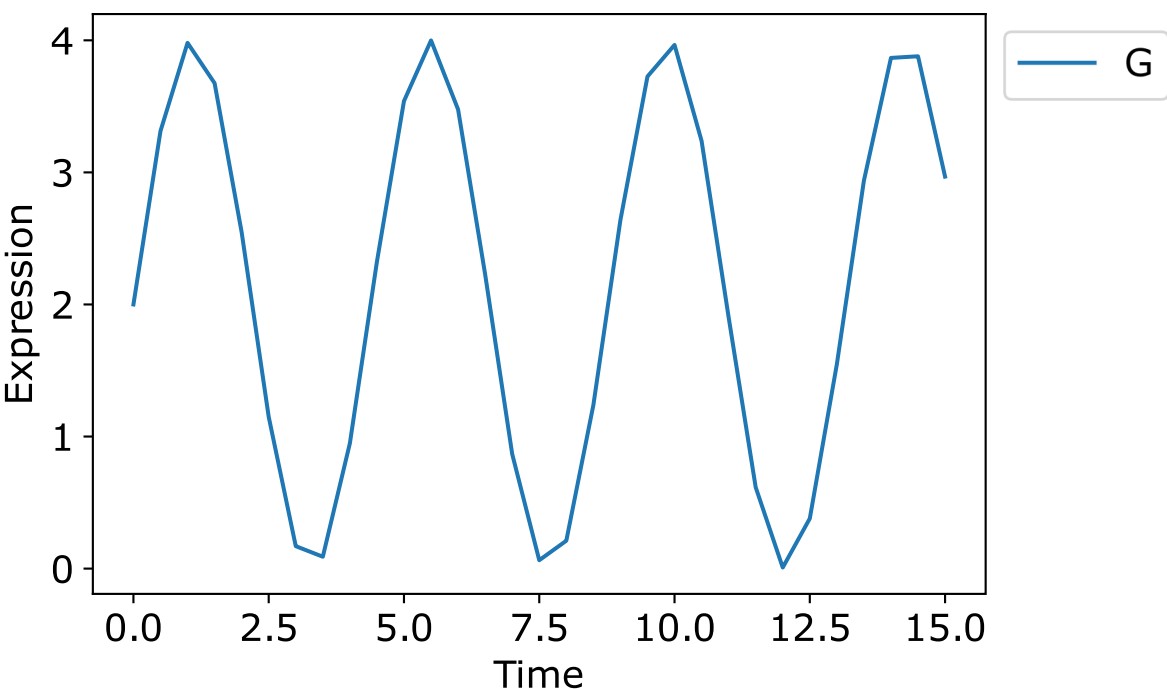

**Fig 8. The true negative time series G for the synthetic network.**

$$\dot{A} = -A + \Gamma_{F,A} + \Omega_{F,A}\frac{F^n}{F^n + \theta_{F,A}^n}$$

$$\dot{B} = -B + \Gamma_{C,B} + \Omega_{C,B}\frac{C^n}{C^n + \theta_{C,B}^n}$$

$$\dot{C} = -C + \Gamma_{A,C} + \Omega_{A,C}\frac{A^n}{A^n + \theta_{A,C}^n} + \Gamma_{E,C} + \Omega_{E,C}\frac{E^n}{E^n + \theta_{E,C}^n}$$

$$\dot{E} = -E + \left(\Gamma_{B,E} + \Omega_{B,E}\frac{\theta_{B,E}^n}{B^n + \theta_{B,E}^n}\right)\left(\Gamma_{F,E} + \Omega_{F,E}\frac{F^n}{F^n + \theta_{F,E}^n}\right)$$

$$\dot{F} = -F + \left(\Gamma_{B,F} + \Omega_{B,F}\frac{\theta_{B,F}^n}{B^n + \theta_{B,F}^n}\right)$$

(5)

Note that the parameter $\Gamma_{D,C}$ is one or two orders of magnitude smaller for Fig 3B than in the parameter sets corresponding to Fig 3D and 3C respectively. In the translation of the DSGRN modeling framework to Hill function ODEs (4), $\Gamma_{D,C}$ distributes to the coefficients on each of the nonlinearities describing activation of node C by E and activation of node C by A. The effect is a more significant reduction in the relative maximum strength of regulation of node C by E and A for the first parameter choice compared to the other two. In other words, D's maximum strength of regulation on C is made comparatively much stronger than the other two inputs to node C in the first parameter (Fig 3B). This may partially explain the observation that node D serves an essential role in maintaining system oscillations in the parameterization producing Fig 3B and 3F, but not in the other parameterizations.

**Table 2. Synthetic network ODE parameters.**

| | DSGRN Parameter Region | | |
|---|---|---|---|
| | **Fig 3B and 3F** | **Fig 3C and 3G** | **Fig 3D and 3H** |
| $\Gamma_{B,F}$ | 0.1071294686 | 0.1247062357 | 0.1054951481 |
| $\Omega_{B,F}$ | 1.1980739800 | 2.6962820818 | 2.0937997708 |
| $\theta_{B,F}$ | 0.0055430383 | 0.3637389501 | 0.5673228474 |
| $\Gamma_{F,E}$ | 0.0175034071 | 0.5011731410 | 0.2455250910 |
| $\Omega_{F,E}$ | 0.8244263241 | 0.1209518139 | 1.5683516101 |
| $\theta_{F,E}$ | 1.3029576011 | 1.5201526697 | 0.8518129907 |
| $\Gamma_{C,B}$ | 0.0011155796 | 0.1958127811 | 0.0966678191 |
| $\Omega_{C,B}$ | 1.9491367094 | 1.5533875523 | 2.3227225179 |
| $\theta_{C,B}$ | 2.9810640658 | 3.5689954379 | 3.1482725035 |
| $\Gamma_{E,D}$ | 0.2201158863 | 0.2141223259 | 0.0529032838 |
| $\Omega_{E,D}$ | 3.6625872030 | 2.9847903373 | 0.5576409019 |
| $\theta_{E,D}$ | 0.5390177462 | 1.0533043028 | 0.3727988706 |
| $\Gamma_{E,C}$ | 1.6499211746 | 0.4358121924 | 0.0676798151 |
| $\Omega_{E,C}$ | 0.3900059619 | 0.5490600005 | 1.7873719546 |
| $\theta_{E,C}$ | 0.0088027178 | 1.4982296668 | 1.4187706085 |
| $\Gamma_{A,C}$ | 0.1007846993 | 0.4132072556 | 0.0928646536 |
| $\Omega_{A,C}$ | 0.2578132626 | 2.2969011759 | 1.2197732691 |
| $\theta_{A,C}$ | 0.2419230934 | 0.6819105357 | 1.7352137828 |
| $\Gamma_{B,E}$ | 0.0096494654 | 0.2810071904 | 0.6185173324 |
| $\Omega_{B,E}$ | 0.7130182868 | 2.2758756351 | 3.7129348824 |
| $\theta_{B,E}$ | 1.5106988697 | 0.6517967757 | 0.7417231063 |
| $\Gamma_{A,D}$ | 0.0542085170 | 0.1210124702 | 0.2731054750 |
| $\Omega_{A,D}$ | 0.8946601377 | 2.0147888846 | 0.5576412929 |
| $\theta_{A,D}$ | 0.6862437739 | 1.1760092769 | 0.4867821250 |
| $\Gamma_{D,C}$ | 0.0624991995 | 1.1124366444 | 0.2593204887 |
| $\Omega_{D,C}$ | 1.3297386450 | 3.2764296326 | 4.0461307096 |
| $\theta_{D,C}$ | 1.0817547392 | 2.1191576389 | 0.1931413047 |
| $\Gamma_{F,A}$ | 0.1885194202 | 0.0567438444 | 0.0398781756 |
| $\Omega_{F,A}$ | 1.4488910403 | 2.1341889294 | 6.1842079625 |
| $\theta_{F,A}$ | 0.7285164419 | 0.3815742876 | 1.4900313815 |
| n | 5 | 5 | 5 |

**4.4.2 Synthetic network hyperparameters.** For each target/regulator pair, denoted ($A$, $B$), LEM first optimizes the choice of model parameters $\Gamma = (\alpha, \beta, \gamma, k, n)$, from Methods Section 4.2.2, applied to the right hand side of the Hill model ODE $\frac{dA}{dt} = f(B; \Gamma)$. The optimization Python package, scipy.optimize.basinhopping, attempts to find the choice of $\Gamma$ which globally minimizes the loss (mean square error) between the model prediction and the measured target time series. This is done by repeatedly running local optimizations (# iterations times) at different starting locations determined by random jumps in parameter space with a maximum displacement of size step_size and an accept/reject criterion controlled by the "temperature" hyperparameter, $T$; see Table 3. The "interval" hyperparameter controls the number of iterations between adjustments of the step size hyperparameter. Local optimizations are performed using the bounded, limited memory BroydenFletcherGoldfarbShanno algorithm.

**Table 3. Edge finding hyperparameters for synthetic network and yeast cell cycle applications.**

| Loss function | Local Optimizer | # iterations | T | step_size | interval |
|---|---|---|---|---|---|
| MSE | "L-BFGS-B" | 10 | 1 | 0.5 | 10 |

The local edge ranking of LEM informs the network finding step through the choice of seed network and additional edges to be used for network sampling. A user-chosen cutoff for the LEM score (LEM pld threshold) determines the edges in the seed network, and the user also specifies the number of additional edges to use in network construction. The seed edges and user-specified LEM edges together form the top-ranked LEM edges introduced in Table 1. The network neighborhood search around the seed network is constrained by network finding hyperparameter choices, the most important of which are topological constraints, probabilities for adding or removing nodes and edges, the range of such operations to perform, the noise levels at which to compute the sequence of extrema in the data, and the maximum size of the networks allowed given in terms of the number of DSGRN parameter regions for the network. The maximum size must be limited for computational reasons; the number of DSGRN parameter regions scales combinatorially with the number of edges in the network. In the following, we limited ourselves to networks with at most 3000 DSGRN parameter regions. The network in Fig 3A has 2016 DSGRN parameter regions. As mentioned in the Introduction, we are searching for core oscillator behavior, or strongly periodic signals that drive large-scale downstream oscillations. A key assumption is that a core oscillator is strongly connected, i.e. that there is a feedback path from every node to every other node in the network. We only sampled networks in the network finding step with this topological property.

The remainder of the network finding hyperparameters are shown in Table 4. Regarding noise levels, since we have reasonably smooth data we evaluated the sequence of extrema at a 0% noise level. For experimental data, this is not a reasonable choice, and a nonzero value will be chosen in our demonstration of the yeast cell cycle. The choice of seed network, the number of user-specified LEM edges, and the probabilities of adding and removing nodes and edges all depend on the level of trust in LEM output. For the synthetic network, we decided to put absolute trust in the very top LEM scores, but weak trust that all relevant network edges are highly ranked. In particular, we chose a seed network composed of all LEM edges with a probability greater than 0.98 and permitted only the addition of nodes and edges. We chose to explore a neighborhood of the seed network that permits a range of 2–10 additions from the next 40 (of 98 total) LEM-ranked edges, excluding self-repressing edges as mentioned earlier.

In Fig 4, networks derived from the well ranked LEM edges (Fig 4A) show the oscillation and pattern match scores for one of the five simulations for the parameterization in Fig 3B. Networks derived from the poorly ranked LEM edges (Fig 4B) use the bottom 50 ranked edges in the same edge finding step along with an empty seed network to repeat the network finding step with the same hyperparameters.

**Table 4. Network finding hyperparameters for synthetic network.**

| LEM pld threshold | user-specified LEM edges | number of operations | noise level |
|---|---|---|---|
| 0.98 | 40 | 2–10 | 0% |
| prob. add node | prob. add edge | prob. drop node | prob. drop edge |
| 0.1 | 0.9 | 0.0 | 0.0 |

**Table 5. The list of 9 substantiated and 2 unsubstantiated yeast cell cycle genes.**

| Substantiated Nodes | Unsubstantiated Nodes |
|---|---|
| ASH1 | EDS1 |
| CLN3 | RIF1 |
| HCM1 | |
| NDD1 | |
| NRM1 | |
| SWI4 | |
| SWI5 | |
| WHI5 | |
| YOX1 | |

**4.4.3 Yeast cell cycle hyperparameters.** Table 5 lists the genes that were investigated in the yeast cell cycle study. The nine genes on the left are known to participate in the yeast cell cycle through extensive experimentation [34, 35, 38], where we have chosen to focus on transcription factors with the addition of only one protein-protein mediated regulator, CLN3. The two genes on the right have not been implicated in the yeast cell cycle, and yet are transcription factors that exhibit high amplitude, robust periodicity of the same period as the cell cycle according to DL×JTK analysis [25].

To choose "true positive" regulatory edges, we use YEASTRACT, a database that compiles experimental evidence for regulatory interactions in the yeast genome [7]. We assume that regulatory edges that have documented transcriptional evidence in YEASTRACT are "ground truth", or substantiated edges. When using YEASTRACT, we specified that expression level data was required for a regulatory interaction; binding-only relationships were not used. We augment the substantiated edges list by the three interactions WHI5 repressed by CLN3, SWI4 repressed by WHI5, and SWI4 repressed by NRM1 from the cell cycle network model in [35]. The full list of 24 substantiated edges is in Table 6. All other putative regulatory edges are deemed unsubstantiated. These include all regulatory interactions between RIF1 and EDS1 with SWI4, NDD1, SWI5, HCM1, CLN3, WHI5, NRM1, YOX1, or ASH1, none of which were found in YEASTRACT.

The time series input into the Inherent dynamics pipeline are two replicates of wild type *S. cerevisiae* grown in standard media with microarray time series collected and processed in [38]. We dropped the first time point in each of the replicate time series to remove the stress response from the synchronization via centrifugal elutriation.

The edge finding hyperparameters are given in Table 3, and do not differ from those chosen for the synthetic network. However, the network finding hyperparameters have substantially changed. Referencing Table 7, the number of top-ranked LEM edges has increased from 40 up to 75, in addition to the seed network edges. This is due to the increased number of edges analyzed, 108–242 depending on scenario, instead of 98, which is due to the increased number of nodes and the presence or absence of nontrivial annotations. Also due to the increased number of edges, we increased the network sample size from 2000 to 4000 and correspondingly decreased the number of DSGRN parameter regions from 3000 to 2000 for computational reasons. Another change is that nonzero probabilities for node and edge removal have been specified. This represents a decrease in the trust of top rankings in LEM, simply because the data are noisier. For the same reason, a noise level of 5% instead of 0% was chosen for analyzing the time series.

**Table 6. Substantiated edges used in the yeast cell cycle results as determined from YEASTRACT and [35].** Every node acts both as a target and as a source. When annotations are specified, HCM1, NDD1, and SWI5 are activators only and NRM1, CLN3, and WHI5 are repressors only, as can be verified from the table. For example, HCM1 activates NDD1, NRM1, and WHI5, but has no repressing activity. All other nodes may be either activators or repressors.

| Target | Regulation | Source |
|--------|-----------|--------|
| ASH1 | act by | SWI5 |
| ASH1 | rep by | YOX1 |
| CLN3 | act by | SWI5 |
| CLN3 | rep by | SWI4 |
| CLN3 | rep by | YOX1 |
| HCM1 | act by | SWI4 |
| HCM1 | rep by | YOX1 |
| HCM1 | rep by | ASH1 |
| NDD1 | act by | SWI4 |
| NDD1 | act by | HCM1 |
| NRM1 | act by | SWI4 |
| NRM1 | act by | HCM1 |
| NRM1 | rep by | YOX1 |
| SWI4 | act by | SWI4 |
| SWI4 | rep by | WHI5 |
| SWI4 | rep by | NRM1 |
| SWI4 | rep by | YOX1 |
| SWI5 | act by | NDD1 |
| SWI5 | rep by | YOX1 |
| WHI5 | act by | HCM1 |
| WHI5 | rep by | CLN3 |
| YOX1 | act by | SWI4 |
| YOX1 | act by | YOX1 |
| YOX1 | act by | ASH1 |

**Table 7. Network finding hyperparameters for yeast cell cycle.**

| LEM pld threshold | user-specified LEM edges | number of operations | noise level |
|-------------------|--------------------------|----------------------|-------------|
| 0.98 | 75 | 2–10 | 5% |
| prob. add node | prob. add edge | prob. drop node | prob. drop edge |
| 0.1 | 0.6 | 0.1 | 0.2 |

## Supporting information

**S1 Table. Synthetic network results for edge finding.**
(PDF)

**S2 Table. Synthetic network results for network finding.**
(PDF)

**S3 Table. Median edge rankings and average edge prevalence scores over five computations for Fig 3B.**
(PDF)

**S4 Table. Median edge rankings and average edge prevalence scores over five computations for Fig 3C.**
(PDF)

**S5 Table. Median edge rankings and average edge prevalence scores over five computations for Fig 3D.**
(PDF)

**S6 Table. Yeast cell cycle results for edge finding.**
(PDF)

**S7 Table. Yeast cell cycle results for network finding.**
(PDF)

**S1 File. Hill model simulations for a top-ranked yeast cell cycle network.**
(PDF)

## Author Contributions

**Conceptualization:** Breschine Cummins, Francis C. Motta, Robert C. Moseley, Anastasia Deckard, Marcio Gameiro, Tomáš Gedeon, Konstantin Mischaikow, Steven B. Haase.

**Data curation:** Robert C. Moseley.

**Formal analysis:** Breschine Cummins, Francis C. Motta, Robert C. Moseley.

**Funding acquisition:** Tomáš Gedeon, Konstantin Mischaikow, Steven B. Haase.

**Investigation:** Breschine Cummins, Francis C. Motta, Robert C. Moseley.

**Methodology:** Breschine Cummins, Francis C. Motta, Robert C. Moseley, Anastasia Deckard, Tomáš Gedeon, Konstantin Mischaikow, Steven B. Haase.

**Software:** Breschine Cummins, Francis C. Motta, Robert C. Moseley, Anastasia Deckard, Sophia Campione, Marcio Gameiro.

**Visualization:** Breschine Cummins, Robert C. Moseley, Sophia Campione.

**Writing – original draft:** Breschine Cummins, Francis C. Motta, Robert C. Moseley, Steven B. Haase.

**Writing – review & editing:** Breschine Cummins, Francis C. Motta, Tomáš Gedeon, Konstantin Mischaikow, Steven B. Haase.

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
