## [Decision Letter · Decision Letter 0]

13 Jun 2022

Dear Dr. Cummins,

Thank you very much for submitting your manuscript "Experimental Guidance for Discovering Genetic Networks through Iterative Hypothesis Reduction on Time Series" for consideration at PLOS Computational Biology.

As with all papers reviewed by the journal, your manuscript was reviewed by members of the editorial board and by several independent reviewers. In light of the reviews (below this email), we would like to invite the resubmission of a significantly-revised version that takes into account the reviewers' comments.

We cannot make any decision about publication until we have seen the revised manuscript and your response to the reviewers' comments. Your revised manuscript is also likely to be sent to reviewers for further evaluation.

Sincerely,

Attila Csikász-Nagy

Associate Editor

PLOS Computational Biology

William Noble

Deputy Editor

PLOS Computational Biology

Reviewer's Responses to Questions

**Comments to the Authors:**

Reviewer #1: In “Experimental Guidance for Discovering Genetic Networks through Iterative Hypothesis Reduction on Time Series”, the authors propose a computational pipeline to identify potential experimental targets for genetic network structure inference. The pipeline predicts a list of genes, edges, and networks that are essential to network dynamics using a set of algorithms including DLJTK, LEM, and DSGRN, and the method was tested on a six-node synthetic network and the S. cerevisiae cell cycle network.

This paper addresses an interesting topic: how to effectively reduce the largely intractable hypothesis space for network inference problem. In most cases, the number of possible networks far exceeds the dimensions of available data, making experiment prioritization a critical issue. There is definitely room for improvement in the study design and the description of computational algorithms. I have listed below several detailed comments which I hope will be helpful for the authors:

1. I don’t think the title really matches the content of the manuscript. The term "iterative hypothesis reduction" was used by the authors, but I can't find any iterative experiments conducted in this paper. Given the core nodes, interactions, or networks identified in the first round, I'm curious how the authors might design the next round of experiments. The synthetic network could be used to demonstrate how to further reduce the number of possible networks.

2. The authors have implemented a software named DSGRN in the Inherent pipeline. In the original paper [42], only small-scale networks with a maximum of 5 nodes were tested. I would recommend the authors benchmark its performance on larger networks in the paper (even in a SI figure/table). I would also recommend the authors pick one or two of the top-ranked DSGRN, run ODE simulations (or even discrete simulations), and show that they are able to reproduce the target dynamic behaviors.

3. In figure 1, the rightmost part is confusing. The gray arrows point to the minimum of the blue and green curves, but not the red ones. How is it related to the oscillation score or pattern matching score?

4. What does a 100% oscillation score mean (page 9, line 227)? Do the authors imply that the network could oscillate with any parameter set?

5. I am not sure I fully understand Table B.1-B.2 (Table B.1 is not referenced in the main text). I'm curious as to how the authors compare their findings with other algorithms.

6. The authors have claimed that “statistics of the top ranked networks be used to provide a prioritization of experimental interventions” (page 17, line 426; page 22, line 601). I don’t think experimental interventions can be easily conducted at the network level. The authors need to clarify how their pipeline output could be related to a feasible intervention experiment.

Minor comments:

1. Page 3 line 97, “the dominant edges provide predictions of the impact of the interventions” is confusing.

I sincerely hope my comments and suggestions will help you in your work.

Reviewer #2: In this manuscript, Cummins and colleagues report a computational method called Inherent Dynamics Pipeline for selecting small core regulatory networks from large genome-wide datasets. They showcase the performance of their approach using synthetic data as well as microarray data of cell-cycle synchronised budding yeast cultures. My background is in transcriptomics analysis of gene expression and I know little about the specifics of network inference and this is reflected in my review. This being said this analysis seems very thorough and I believe the authors that IDP will be a useful approach to prioritise targets for subsequent validation.

Major:

1) Something I could not really get a sense off is how much IDP relies on the quality of the datasets used as inputs. Measurements and synchronisation noise has always been a big limitation for identifying true periodic genes in transcriptomics data. I am curious of the authors’ opinion on this. Have they tried to increase or decrease measurement noise in their synthetic data? If yes, which part of the pipeline and its predictions was the most affected?

2) It is unclear to me from the budding yeast analysis presented in this paper how good IDP would be at identifying previously unknown regulatory nodes. Did I miss it? Could the authors comment please?

Minor:

3) Figure 1: I find it a bit difficult to relate the diagrammes shown on the figure with the node/edges finding process. Maybe an extended legend could help?

4) P4 L117: “Experimental approaches to accomplish this identification task are largely intractable.”. What do the authors mean by intractable here.

5) P5, L125: “top of the list is enriched for core regulatory elements most critical to controlling oscillatory dynamics”. I don’t understand why regulatory elements are expected to be themselves oscillating? Their activation could be post-transcriptional, which is quite commonly seen in the case of cell-cycle regulation. Could the authors elaborate on this please?

6) Figure 2. It would be nice to have a visual representation of the results in the main paper not just in the Appendix.

7) P9, L207 and Figure 3: This section is unclear. What are the implications of this identifiability problem for the users?

8) P10, L235: “…due simply to spurious correlations in their transcriptional…”. What do the authors mean by spurious?

9) P10, L246: The discussion of condition specific node is interesting. However, I am not too sure how this relates to inference by the ID pipeline. Could the author clarify how their analysis presented on panels e-h affect the use of their pipeline please?

Reviewer #3: Review uploaded as attachment

**Have the authors made all data and (if applicable) computational code underlying the findings in their manuscript fully available?**

Reviewer #1: None

Reviewer #2: Yes

Reviewer #3: Yes

PLOS authors have the option to publish the peer review history of their article (what does this mean?). If published, this will include your full peer review and any attached files.

Reviewer #1: No

Reviewer #2: No

Reviewer #3: **Yes: **Marc D Ruben
---

## [Decision Letter · Decision Letter 1]

5 Sep 2022

Dear Dr. Cummins,

We are pleased to inform you that your manuscript 'Experimental Guidance for Discovering Genetic Networks through Hypothesis Reduction on Time Series' has been provisionally accepted for publication in PLOS Computational Biology.

Best regards,

Attila Csikász-Nagy

Academic Editor

PLOS Computational Biology

William Noble

Section Editor

PLOS Computational Biology

Reviewer's Responses to Questions

**Comments to the Authors:**

Reviewer #1: I am satisfied with the revised manuscript and I have no further major or minor comments.

Reviewer #2: I thank the authors for their reply. I have no further concern. If anything, regrading point 1, the authors provided a nice discussion of the impact of data quality on different parts of their pipeline. I would recommend, space permitting, that this is included in the final manuscript so that future readers can benefit from it too.

Reviewer #3: The authors have addressed my concerns and should be commended for a very nice manuscript!

**Have the authors made all data and (if applicable) computational code underlying the findings in their manuscript fully available?**

Reviewer #1: None

Reviewer #2: None

Reviewer #3: Yes

PLOS authors have the option to publish the peer review history of their article (what does this mean?). If published, this will include your full peer review and any attached files.

Reviewer #1: No

Reviewer #2: No

Reviewer #3: **Yes: **Marc D Ruben

---

## [Editor Report · Acceptance letter]

4 Oct 2022

PCOMPBIOL-D-22-00666R1 

Experimental Guidance for Discovering Genetic Networks through Hypothesis Reduction on Time Series

Dear Dr Cummins,

I am pleased to inform you that your manuscript has been formally accepted for publication in PLOS Computational Biology. Your manuscript is now with our production department and you will be notified of the publication date in due course.

With kind regards,

Zsofia Freund
